# Anti-Adipogenic Activity of Secondary Metabolites Isolated from *Smilax sieboldii* Miq. on 3T3-L1 Adipocytes

**DOI:** 10.3390/ijms24108866

**Published:** 2023-05-16

**Authors:** Yeon Woo Jung, Jung A Lee, Ji Eun Lee, Hanna Cha, Yun-Hyeok Choi, Wonsik Jeong, Chun Whan Choi, Joa Sub Oh, Eun-Kyung Ahn, Seong Su Hong

**Affiliations:** 1Bio-Center, Gyeonggido Business & Science Accelerator, Suwon 16229, Republic of Korea; jion123@gbsa.or.kr (Y.W.J.); lovelee90@gbsa.or.kr (J.A.L.); jieun@gbsa.or.kr (J.E.L.); palansky317@gbsa.or.kr (H.C.); choiyh1400@gbsa.or.kr (Y.-H.C.); ws2009@gbsa.or.kr (W.J.); cwchoi@gbsa.or.kr (C.W.C.); 2College of Pharmacy, Dankook University, Cheonan 31116, Republic of Korea; jsoh@dankook.ac.kr

**Keywords:** *Smilax sieboldii*, Smilacaceae, phenylpropanoid glyceride, gallotannin, lanostane triterpenoid, adipogenesis

## Abstract

*Smilax sieboldii*, a climbing tree belonging to Smilacaceae, has been used in traditional oriental medicine for treating arthritis, tumors, leprosy, psoriasis, and lumbago. To evaluate the anti-obesity effects of *S. sieboldii* (Smilacaceae), we screened methylene chloride (CH_2_Cl_2_), ethyl acetate (EtOAc), aqueous-saturated *n*-butanol, and ethanol (EtOH) extracts of the whole plant at various concentrations to inhibit adipogenesis in adipocytes. The 3T3-L1 cell line with Oil red O staining with the help of fluorometry was used as an indicator of anti-obesity activity. Bioactivity-guided fractionation of the EtOH extract and subsequent phytochemical investigation of the active CH_2_Cl_2_- and EtOAc-soluble fractions resulted in the isolation of 19 secondary metabolites (**1**–**19**), including a new α-hydroxy acid derivative (**16**) and two new lanostane-type triterpenoids (**17** and **18**). The structures of these compounds were characterized using various spectroscopic methods. All the isolated compounds were screened for adipogenesis inhibition at a concentration of 100 μM. Of these, compounds **1**, **2**, **4**–**9**, **15**, and **19** significantly reduced fat accumulation in 3T3-L1 adipocytes, especially compounds **4**, **7**, **9**, and **19**, showing 37.05 ± 0.95, 8.60 ± 0.41 15.82 ± 1.23, and 17.73 ± 1.28% lipid content, respectively, at a concentration of 100 μM. These findings provide experimental evidence that isolates from *S. sieboldii* extracts exert beneficial effects regarding the regulation of adipocyte differentiation.

## 1. Introduction

Obesity, the most prevalent chronic metabolic condition, is a state associated with excess body fat accumulation caused by a negative energy balance between caloric intake and expenditure, with more than 1.9 billion overweight adults and 650 million clinically obese people worldwide [1]. Obesity is a significant risk factor for a number of comorbid illnesses, including cardiovascular disease, certain types of cancer, cerebrovascular incidents, metabolic syndrome, nonalcoholic fatty liver disease, obstructive sleep apnea, osteoarthritis, psychiatric troubles, respiratory problems, and type 2 diabetes mellitus, amongst many others [2,3,4]. According to the guidelines for the treatment of obesity, the most effective method for weight management is a multidisciplinary approach that may include behavioral therapy, medication, lifestyle adjustments, and/or bariatric surgery [5]. These strategies include changes in lifestyle (such as diet, exercise, and behavioral therapy), drug therapies (such as anti-obesity drugs that target appetite, absorption, or metabolism), and surgical interventions (such as bariatric surgery that modifies the anatomy or function of the gastrointestinal tract). However, these treatments have drawbacks and obstacles, including low efficacy, adverse effects, high cost, limited accessibility, low adherence, and a high recurrence rate. Therefore, more effective and individualized approaches are required to treat obesity and its comorbidities [2,6]. Generally, both hypertrophy (enhancing the size of existing adipocyte cells) and hyperplasia (boosting the quantity by differentiation of new adipocytes) of fat-storing cells are considered therapeutic targets for obesity treatment because they contribute to the abnormal expansion of adipocytes that is a characteristic of obesity [7,8,9]. Adipose tissue masses are controllable by adipogenesis inhibition, a process that turns fibroblastic preadipocytes into mature fat cells [10]. In vitro, 3T3-L1 preadipocytes, which are mouse embryonic fibroblasts generated cell lines, differentiate into mature adipocyte-like cells, and intracellular buildup of lipid droplets is seen during cell differentiation [11]. Thus, this cell line is a suitable screening model for obesity-related adipose tissue biology. Some natural products and plant extracts, such as *Abelmoschus manihot* [12], *Albizia julibrissin* [13], *Amomum tsao-ko* [14], indole derivatives [15], (−)-loliolide [16], kaempferol [17], Salix pseudolasiogyne [18], and shrimp oil [19], are known to prevent the lipid accumulation or the differentiation of adipocytes in 3T3-L1 cells.

Plants of the *Smilax* (Smilacaceae) genus, commonly called sarsaparilla, consist of approximately 300–350 species. The *Smilax* species, which are widely dispersed throughout the world’s tropical regions and the temperate zones of North America and East Asia, were commonly used as food and traditional medicine to cure inflammatory illnesses [20,21]. *Smilax sieboldii* Miq. is a climbing plant with prickly stems that grows in Korea, Japan, China, and Taiwan. Young leaves are harvested from the wild for local use as food. In addition, the subterranean parts have been employed in traditional folk remedies for arthritis, tumors, leprosy, psoriasis, and lumbago [22]. *S. sieboldii* exhibits various pharmacological activities, including antihyperlipidemic effects and cAMP phosphodiesterase inhibition [23,24]. Phytochemical studies revealed that steroids (spirostane and furostane skeleton) and their corresponding glycosides were present in the rhizome of this plant [22,24] and had several biological activities, such as antifungal, cytotoxic, anti-inflammatory, and anti-bacterial [20,21,25,26]. However, only a few studies have been conducted on the chemical composition of *S. sieboldii*. In our present research, we conducted investigations to identify potential bioactive phytochemicals in the ethyl acetate (EtOAc) and methylene chloride (CH_2_Cl_2_) fraction of the *S. sieboldii* EtOH extract. We included chromatographic purification and structural identification of one undescribed α-hydroxy acid derivative (**16**) and two new lanostane-type triterpenoids (**17** and **18**), which were isolated along with additional known constituents (**1**–**15** and **19**). The structure of the novel constituents was characterized using various spectral experiments (1D/2D NMR and HR-MS) and specific optical rotation analyses. Furthermore, we reported the isolation and structural elucidation of compounds **1**–**19** and evaluated the inhibitory effects of these isolated components on adipocyte differentiation using 3T3-L1 cells.

## 2. Results and Discussion

### 2.1. Isolation and Purification of Compounds ***1***–***19***

Dried whole *S. sieboldii* plant was crushed and extracted with 70% EtOH at room temperature to obtain the crude EtOH extract using rotary evaporation. The EtOH extract was sequentially employed in the solvent partition process with three solvents, CH_2_Cl_2_, EtOAc, and aqueous-saturated *n*-butanol (BuOH), which yielded three main solvent fractions with increasing polarity. To determine the cell viability, an MTT assay was performed by treating 3T3-L1 cells with crude extract (50, 100, and 150 μg/mL). As shown in Appendix A, crude extract showed no significant adverse effect on viability after 24 h, indicating a noncytotoxic effect of crude extract on 3T3-L1 cells. The crude extracts and solvent layers of *S. sieboldii* whole plant were screened for inhibition of adipocyte differentiation at various concentrations (ranging from 3.125 to 150 μg/mL, Figure 1). The CH_2_Cl_2_^−^ and EtOAc-layers showed a decrease in lipid accumulation in low-concentration (Figure 1). Chromatographic purification of the CH_2_Cl_2_^−^ and EtOAc-soluble fractions afforded one new α-hydroxy acid derivative (**16**) and two new lanostane-type triterpenoids (**17** and **18**), along with 16 known compounds (Figure 2).

### 2.2. Chemical Identification of Compounds ***1***–***19***

Compound **16** was isolated as a white amorphous powder with optical rotation of ${\left[\alpha \right]}_{D}^{21}$ −4.8° (c = 0.14, MeOH). The molecular formula of C_7_H_12_O_5_ was established from its high-resolution ESI-MS analysis (*m*/*z* 199.0575 [M + Na]^+^, calculated for C_7_H_12_O_5_Na: 199.0577), indicating two degrees of unsaturation. ^1^H NMR and distortionless enhancement by polarization transfer (DEPT) spectra in combination with the heteronuclear single quantum coherence (HSQC) spectrum revealed characteristic signals of the malic acid moiety [δ_H_ 4.16 (1H, dd, *J* = 8.4, 4.2 Hz), 2.71 (1H, d, *J* = 16.1, 4.2 Hz), and 2.60 (1H, d, *J* = 16.1, 8.4 Hz); δ_C_ 174.7 (s), 173.5 (s), 78.6 (d), and 39.4 (t)] [27,28]. In addition, the ^1^H and ^13^C NMR data of **16** showed the presence of a methoxyl group [δ_H_ 3.41 (3H, s); δ_C_ 59.0 (q)], one oxygenated methylene signals [δ_H_ 4.22 (1H, dd, *J* = 7.0, 2.8 Hz) and 4.20 (1H, dd, *J* = 7.0, 3.5 Hz); δ_C_ 62.4 (t)], and a methyl group [δ_H_ 1.28 (3H, t, *J* = 7.0 Hz); δ_C_ 14.6 (q)] (Table 1, Appendix A). Spin–spin coupling was observed as a cross-peak between the oxygenated methylene signals (δ_H_ 4.22 and 4.20) and the methyl signal (δ_H_ 1.28) in the ^1^H-^1^H correlation spectroscopy (COSY) spectrum, representing the ethoxy group (Figure 3). In the heteronuclear multiple bond correlation (HMBC) spectrum, the correlations between the methoxy group proton at δ_H_ 3.41 and C-2 (δ_C_ 78.6) confirmed that the methoxy group was linked to C-2, and the HMBC correlation of the oxygenated methylene protons (δ_H_ 4.22 and 4.20) and C-1 (δ_C_ 173.5) indicated that the ethoxy group was connected at C-1. Based on the above results, the structure of **16** was identified as 1-ethyl-2-methoxy malate and was named sieboldic acid.

Compound **17** was isolated as white amorphous powder and displayed a positive Liebermann–Burchard reaction. Its molecular formula was determined to be C_32_H_54_O from the molecular ion at *m*/*z* 455.4275 [M + H]^+^ based on HR-ESIMS, indicating six indices of hydrogen deficiency. The IR spectrum showed absorption bands at hydroxyl (3382 cm^−1^) and terminal double bond (3043, 1371, and 890 cm^−1^) functionalities [29]. The ^1^H-NMR spectrum of **17** (Table 2) indicated the presence of eight tertiary methyls [δ_H_ 1.03, 0.97, 0.80, 0.73, 0.64 (3H each, s), and 1.04 (3H × 3, s)], one secondary methyl [δ_H_ 0.90 (3H, d, *J* = 6.3 Hz)], one axial methine proton bearing a hydroxy [δ_H_ 3.20 (1H, dd, *J* = 11.2, 4.2 Hz)], two germinal olefinic protons [δ_H_ 4.82, 4.65 (1H each, s)], and a typical H-11 proton of lanosta-9(11)-ene [δ_H_ 5.21 (1H, d, *J* = 5.6 Hz)]. The ^13^C-NMR spectrum of **17** revealed 32 carbon signals, which were identified with the aid of a DEPT experiment as four methines (δ_C_ 52.7, 51.2, 42.0, and 36.71), one oxygenated methine (δ_C_ 79.1), nine methylenes (δ_C_ 37.4, 36.69, 36.3, 34.1, 28.4, 28.3, 28.2, 28.0, and 21.6), nine methyls (δ_C_ 29.6 × 3, 28.5, 22.5, 18.73, 18.71, 15.9, and 14.6), five quaternary carbons (δ_C_ 47.2, 44.5, 39.6, 39.3, and 36.5), and four olefinic carbons (δ_C_ 159.2, 148.7, 115.2, and 105.9). Based on these NMR features and a comparison of ^1^H and ^13^C-NMR data for **17** and agrostophyllinol, isolated from *Agrostophyllum brevipes* [30], it was confirmed that these two compounds have a same plane skeleton [lanost-9(11)-ene skeleton, observed in C-3, C-5, C-9, C-10, C-11, C-13, C-14, C-17, C-18, C-21, and C-30 (∆δ_C_ +0.2, +0.2, +0.2, +0.6, +0.3, +0.2, +0.2, +0.3, +0.3, +0.3, +0.4 (∆δ_C_ = δ**_17_** − δ_lanost-9(11)-ene_))]. Moreover, the HMBC correlations of the five individual tertiary methyl signals on rings A-D [between CH_3_-18 (δ_H_ 0.64) and C-12 (δ_C_ 37.4), C-13 (δ_C_ 44.5), C-14 (δ_C_ 47.2), and C-17 (δ_C_ 51.2); between CH_3_-19 (δ_H_ 1.03) and C-1 (δ_C_ 36.3), C-5 (δ_C_ 52.7), C-9 (δ_C_ 148.7), and C-10 (δ_C_ 39.6); between CH_3_-28 (δ_H_ 0.97) and C-4 (δ_C_ 39.3), C-3 (δ_C_ 79.1), C-5, and C-29 (δ_C_ 15.9); between CH_3_-29 (δ_H_ 0.80) and C-3, C-4, C-5, and C-28 (δ_C_ 28.5); between CH_3_-30 (δ_H_ 0.73) and C-8 (δ_C_ 42.0), C-13, C-14, and C-15 (δ_C_ 34.1)] and HMBC cross-peaks between H-11 (δ_H_ 5.21) and C-8, C-9, C-10, C-12, and C-13 firmly established the linkages of these partial structural units (Figure 4). The only distinction between them was the presence of an additional methyl group at C-25 of the side chain in compound **17**. This was supported by the singlet signal of three chemically equivalent methyl groups [δ_H_ 1.04 (3H × 3, s, H-26, H-27, H-32)] observed in the ^1^H-NMR spectrum of **17**, which revealed the HMBC correlations with C-24 (δ_C_ 159.2) and ROESY correlations with H-31 (δ_H_ 4.82), elucidating the structure of the side chain as CH(CH_3_)CH_2_CH_2_C(CH_2_)C(CH_3_)_3_ (Figure 4). The relative configuration for **17** was determined as follows: first, the large coupling constant of H-3 (*J*_2–3_ = 11.2, 4.2 Hz) indicated that the hydroxyl group (3-OH) was oriented equatorially (β-position) at C-3 [29,31]. Next, the ROESY correlations between H-18 (δ_H_ 0.64) and H-20 (δ_H_ 1.39); and H-17 (δ_H_ 1.61) and H-21 (δ_H_ 0.90) were determined as *R* configuration for the stereochemistry of C-20. Thus, compound **17** was accordingly determined as 25-methyl-24-methylenelanost-9(11)-en-3β-ol and was given the trivial name smilaxsiebolane A.

Compound **18** was obtained as white amorphous powder and exhibited a deprotonated molecular ion peak at *m*/*z* 453.4070 [M + H]^+^ (calcd for C_32_H_53_O, *m*/*z* 453.4091) on the HRESI-MS analysis, indicating its elemental formula as C_32_H_52_O. The molecular formula revealed seven indices of hydrogen deficiency. The following ^1^H-NMR and HSQC analysis revealed one olefin proton [δ_H_ 5.29 (1H, d, *J* = 6.3 Hz)], eight tertiary methyls [δ_H_ 1.23, 1.08, 1.07, 0.75, 0.68 (3H each, s), and 1.06 (3H × 3, s)], one secondary methyl [δ_H_ 0.93 (3H, d, *J* = 6.3 Hz)], two germinal olefinic protons [δ_H_ 4.84, 4.67 (1H each, s)], and four methines (δ_H_ 2.22, 1.65, 1.43, and 1.37) (Table 2). The ^13^C-NMR data, in accordance with HRESI-MS data, showed 32 carbon signals, including one ketone carbon (δ_C_ 217.2), nine methyls (δ_C_ 29.3 × 3, 25.6, 22.0, 21.8, 18.5, 18.4, and 14.4), four olefinic carbons (δ_C_ 159.0, 147.1, 116.3, and 105.8), four methine carbons (δ_C_ 53.4, 50.9, 41.9, and 36.5), nine methylenes (δ_C_ 37.2, 36.7, 36.4, 34.9, 33.9, 28.2, 28.0, 27.7, and 22.6), and five quaternary carbons (δ_C_ 47.7, 47.0, 44.3, 39.1, and 36.3) (Table 2). The ^1^H- and ^13^C-NMR spectra of **18** were comparable to those of **17**, with the exception of the presence of a ketone functional group at δ_C_ 217.2 (C-3) and, meanwhile, the disappearance of the H-3 proton signal. As anticipated, the ^1^H NMR spectrum of **18** is absent of the signal at δ_H_ 3.20 (hydroxy methine proton of compound **17**) and instead displays two multiplet protons at δ_H_ 2.72 and 2.40, indicating a ketomethylene group. This assumption was validated by HMBC correlation of H-1 (δ_H_ 2.10 and 1.81), H-2 (δ_H_ 2.72 and 2.40), H-28 (δ_H_ 1.08), and H-29 (δ_H_ 1.07) with C-3 (δ_C_ 217.2) (Figure 5). Compound **18** has the same relative configuration as **17**, confirmed by ROESY data (Figure 5) and positive specific rotation value ${\left[\alpha \right]}_{D}^{25}$ +26.4° (c = 0.36, CH_2_Cl_2_). In conclusion, compound **18** was determined as 25-methyl-24-methylenelanost-9(11)-en-3-one and has been named smilaxsiebolane B.

The 16 known compounds were identified as *trans*-3,3′,5,5′-tetrahydroxy-4-methoxystilbene (**1**) [32], resveratrol (**2**) [33], 4-hydroxybenzoic acid (**3**) [34], 2-*O*-caffeoylglycerol (**4**) [35], 1-*O*-caffeoylglycerol (**5**) [36], juncusyl ester B (**6**) [37], acertannin (**7**) [38], 2-*O*-(*E*)-feruloyl glyceride (**8**) [39], maplexin D (**9**) [40], *trans*-caffeic acid (**10**) [41], syringic acid (**11**) [42], 1-syringoyl-1,2-dihydroxyethane (**12**) [43], 3,5-dihydroxy-4-methoxybenzoic acid (**13**) [44], 1-(3,4,5-trimethoxyphenyl)-1,2,3-propanetriol (**14**) [45], 1-*O*-*trans*-ρ-coumaroylglycerol (**15**) [37], and *trans*-ρ-ethyl coumarate (**19**) [46] by comparing their spectroscopic data with reference values from the previously published literature. To the best of our knowledge, this is the first study to isolate all the compounds from *S. sieboldii*. Seven compounds (**1**, **4**, **7**–**9**, **12**, and **13**) were identified in the Smilacaceae family for the first time.

### 2.3. Anti-Adipogenic Effects of Isolated Compounds on 3T3-L1 Cells

As a first step, to evaluate whether the separated phytochemicals had any influence on cell viability, 3T3-L1 preadipocytes were treated for 24 h with various concentrations (25–100 μM) of the compounds. No cytotoxicity was observed up to 100 μM for all compounds (Appendix A); thus, we treated the cells with phytochemicals at this concentration for further study. The anti-adipogenic effects of 19 compounds isolated from the active CH_2_Cl_2_ and EtOAc layer were screened by assessing fat accumulation in 3T3-L1 cells treated with 100 μM of phytochemicals (Figure 6). Different classes of compounds, such as stilbenoids, phenylpropanoid glycerides, simple phenolics, gallotannins (having a 1,5-anhydro-D-glucitol core), and lanostane-type triterpenoids were tested. Oil Red O staining was used to see how effectively the separated compounds inhibited intracellular lipid accumulation on day eight. Out of the 19 compounds, acertannin (**7**), maplexin D (**9**), and *trans*-ρ-ethyl coumarate (**19**) were the most active, with lipid accumulation percentages of 8.60 ± 0.41, 15.82 ± 1.24, and 17.73 ± 1.28%, respectively, at 100 μM treatment (** *p* < 0.01). In addition, 2-*O*-caffeoylglycerol (**4**), resveratrol (**2**), 1-*O*-caffeoylglycerol (**5**), juncusyl ester B (**6**), 2-*O*-(*E*)-feruloyl glyceride (**8**), 1-*O*-*trans*-ρ-coumaroylglycerol (**15**), and *trans*-3,3′,5,5′-tetrahydroxy-4-methoxystilbene (**1**) were moderately active with lipid accumulation % of 37.05 ± 0.96, 39.21 ± 3.98, 46.35 ± 3.56, 54.94 ± 7.40, 55.96 ± 10.52, 66.64 ± 4.78, and 67.93 ± 5.80%, respectively, at 100 μM (* *p* < 0.05, ** *p* < 0.01).

The structural class had a specific correlation with the inhibition of adipocyte differentiation. In the case of phenylpropanoid derivatives, the activity varied significantly with the replacement for glycerol. The *trans*-caffeic acid (**10**) has a free glycerol group and is negligibly active in preventing fat accumulation in adipocytes (lipid content % of 85.77 ± 17.18% at 100 μM). However, the glycerol substitution (2-*O*-caffeoylglycerol and 1-*O*-caffeoylglycerol) on the free carboxylic acid group of the trans-caffeic acid exhibited active fat accumulation inhibition from *S. sieboldii*, with 37.05 ± 0.96 and 46.35 ± 3.56% lipid accumulation at 100 μM, respectively. Among the isolates, gallotannins (acertannin and maplexin D) exhibited the most potent anti-adipogenesis effects, with an activity of 8.60 ± 0.41 and 15.82 ± 1.24% lipid content at 100 μM, respectively. Recently, the anti-adipogenic activity of a gallotannins mixture from *Mangifera indica* (mango) was demonstrated by the reduced number of lipid droplets and signaling pathway stimulation of the AMP-activated protein kinase (AMPK) [47]. The stilbenoid active compound from *S. sieboldii*, resveratrol (**2**), exhibits dose-dependent anti-adipogenic activity [48,49,50]. Moreover, resveratrol has been used as a positive control in the anti-adipogenic effect research literature [51,52]. Resveratrol stimulates deacetylating enzyme sirtuin-1 (SIRT-1) deacetylase, inhibiting adipogenesis, and adipocyte differentiation. Additionally, it increases adenosine monophosphate-activated protein kinase (AMPK) activity, which inhibits adipogenesis [48]. Based on these previous studies, the anti-obesity characteristics of *trans*-3,3′,5,5′-tetrahydroxy-4-methoxystilbene (**1**) are likely. The most active compounds, 2-*O*-caffeoylglycerol (**4**), acertannin (**7**), and maplexin D (**9**), were studied for dose dependency at concentrations of 25, 50, and 100 μM (Figure 7). According to these results, the stilbenoids (**1** and **2**), phenylpropanoid glycerides (**4**–**6**, **8**, and **15**), and gallotannins (**7** and **9**) isolated from *S. sieboldii* possessed potential anti-obesity effects because of their dose-dependent inhibitory activities against adipocyte differentiation and lipid formation in 3T3-L1 cells.

## 3. Materials and Methods

### 3.1. General Experimental Procedure

Detailed information on the general experimental procedures is provided in the Appendix A.

### 3.2. Source of Plant Material

The whole plant of *S. sieboldii* was collected at Yeoncheon-gun, Gyeonggi-do, Republic of Korea, in August 2021. Botanical identification was performed, and a voucher specimen (G99) was deposited at the Bio-Center, Gyeonggido Business and Science Accelerator (GBSA), Suwon, Republic of Korea.

### 3.3. Extraction and Separation/Compound Isolation

The shade-dried whole plant of *S. sieboldii* (7 kg) was percolated with 70% aqueous EtOH at room temperature. Following evaporation of the solvent under reduced pressure, the residue (395 g) was suspended in water, successively partitioned with CH_2_Cl_2_, EtOAc, and water-saturated *n*-BuOH to afford 86 g, 25.5 g, and 39 g, respectively. Part of the EtOAc-soluble fraction was chromatographed over a Diaion HP-20 resin and eluted with water−MeOH stepwise gradient solvent system (1:0 to 0:1) to yield five fractions (G99A_1_ to G99A_5_). Of these, fraction G99A_2_ (2.5 g) was subjected to MPLC over ODS eluting with a gradient of 10–30% MeOH in H_2_O to give 11 subfractions (G99B_1_ to B_11_). Fraction G99B_1_ (72.7 mg) was separated by preparative high-pressure liquid chromatography (HPLC) (Kromasil 100-5-C18 column; 250 × 21.2 mm; flow rate 10 mL/min; solvent A–0.05% trifluoroacetic acid in the water, solvent B–MeCN; gradient elution, 0 min 10% B to 50 min 15% B, detection at 254 and 350 nm). HPLC separation led to the purification of compounds **12** (2.3 mg, *t*_R_ = 18.3 min), **13** (4.5 mg, *t*_R_ = 19.8 min), and **14** (15.8 mg, *t*_R_ = 21.4 min). Further purification of subfraction G99B_2_ (86.9 mg) using the above HPLC system (gradient elution, 0 min 15% B to 50 min 27% B) resulted in the isolation of compound **3** (15.3 mg, *t*_R_ = 16.9 min). Fractions G99B_4_ (125 mg), G99B_8_ (95 mg), and G99B_10_ (86.7 mg) were separated by preparative HPLC (same conditions as for fraction G99B_2_) to obtain pure compounds **4** (25.8 mg, *t*_R_ = 14.3 min), **5** (31.3 mg, *t*_R_ = 16.9 min), **16** (3.5 mg, *t*_R_ = 22.0 min), **6** (35.9 mg, *t*_R_ = 21.4 min), **15** (4.5 mg, *t*_R_ = 25.5 min), **7** (20.1 mg, *t*_R_ = 16.5 min), and **8** (13.4 mg, *t*_R_ = 22.2 min). Subfraction G99B_6_ (69.7 mg) was isolated using preparative HPLC (gradient elution, 0 min 15% B, 50 min 23% B) to obtain three compounds **9** (1.5 mg, *t*_R_ = 12.8 min), **10** (3.1 mg, *t*_R_ = 18.8 min), and **11** (8.2 mg, *t*_R_ = 19.5 min). G99A_3_ (3.5 mg) was separated using RP-MPLC with MeCN−H_2_O (elution 0 min 10:90 to 110 min 40:60) to obtain eight subfractions (G99C_1_ to C_8_). Subfraction G99C_1_ (125 mg) was purified using preparative HPLC (gradient elution, 0 min 17% B, 30 min 30% B) to purify compound **1** (24.8 mg, *t*_R_ = 32.5 min). Subfraction G99C_7_ (150 mg) and subjected to preparative HPLC (detection 280 and 325 nm, gradient elution, 0 min 25% B, 50 min 33% B) to obtain compound **2** (36.7 mg, *t*_R_ = 27.7 min). The CH_2_Cl_2_-soluble fraction was subjected to silica gel column chromatography (CC) and eluted using a CH_2_Cl_2_-MeOH gradient system (1:0 to 1:1) to yield 12 fractions (G99D_1_ to G99D_12_). Compound **17** (125.1 mg) was isolated from fraction G99D_3_ (1.2 g) by recrystallization in CH_2_Cl_2_-MeOH. G99D_5_ (931 mg) was further chromatographed by silica gel CC with an *n*-hexane-CH_2_Cl_2_ (10:1 to 0:1) mixture to obtain nine fractions (G99E_1_ to G99E_9_). The fraction G99E_8_ (68.9 mg) was isolated by Sephadex LH-20 CC (CH_2_Cl_2_-MeOH, 1:1, isocratic elution) to obtain compounds **18** (15.4 mg) and **19** (7.5 mg). The isolation part of the present research is summarized in Figure 8.

### 3.4. Spectroscopic Data Analysis

NMR spectra were obtained in acetone-*d*_6_, CD_3_OD, and CDCl_3_. NMR data were acquired using a Bruker Ascend III 700 spectrometer (resonance frequency was 700.53 and 176.15 MHz for ^1^H and ^13^C, respectively). The ^1^H and ^13^C NMR spectra are shown in Appendix A.*trans-3,3′,5,5′-Tetrahydroxy-4-methoxystilbene* (**1**): pale brown amorphous powder; UV (MeOH) λ_max_ (log ε) 224 (3.55), 314 (3.75) nm; ^1^H-NMR (700 MHz, acetone-*d*_6_) δ 6.91 (1H, d, *J* = 16.1 Hz, H-8), 6.87 (1H, d, *J* = 16.1 Hz, H-7), 6.64 (2H, s, H-2′, 6′), 6.55 (2H, d, *J* = 2.1 Hz, H-2, 6), 6.30 (1H, d, *J* = 2.1 Hz, H-4), 3.82 (3H, s, 4′-OCH_3_); ^13^C-NMR (175 MHz, acetone-*d*_6_) δ 159.6 (C-3, 5), 151.5 (C-3′, 5′), 140.6 (C-1), 136.3 (C-4′), 134.3 (C-1′), 129.3 (C-8), 128.8 (C-7), 106.8 (C-2′, 6′), 105.9 (C-2, 6), 103.0 (C-4), 60.8 (4′-OCH_3_); ESI-MS (positive ion mode) *m*/*z* 275 [M + H]^+^.*Resveratrol* (**2**): pale brown amorphous powder; UV (MeOH) λ_max_ (log ε) 216 (3.56), 305 (3.93) nm; ^1^H-NMR (700 MHz, CD_3_OD) δ 6.95 (1H, d, *J* = 16.1 Hz, H-8), 6.80 (1H, d, *J* = 16.1 Hz, H-7), 7.35 (2H, d, *J* = 8.4 Hz, H-2′, 6′), 6.76 (2H, d, *J* = 8.4 Hz, H-3′, 5′), 6.44 (2H, d, *J* = 2.1 Hz, H-2, 6), 6.16 (1H, t, *J* = 2.1 Hz, H-4); ^13^C-NMR (175 MHz, CD_3_OD) δ 159.8 (C-3, 5), 158.5 (C-4′), 141.5 (C-1), 130.6 (C-1′), 129.5 (C-8), 128.9 (C-2′, 6′), 127.2 (C-7), 116.2 (C-3′, 5′), 105.9 (C-2, 6), 102.8 (C-4); ESI-MS (positive ion mode) *m*/*z* 229 [M + H]^+^.*4-Hydroxybenzoic acid* (**3**): white amorphous powder; UV (MeOH) λ_max_ (log ε) 199 (3.95), 210 (sh), 255 (3.70) nm; ^1^H-NMR (700 MHz, CD_3_OD) δ 7.88 (2H, d, *J* = 8.4 Hz, H-2, 6), 6.82 (2H, d, *J* = 8.4 Hz, H-3, 5); ^13^C-NMR (175 MHz, CD_3_OD) δ 170.2 (C-7), 163.5 (C-4), 133.1 (C-2, 6), 122.8 (C-1), 116.2 (C-3, 5); ESI-MS (positive ion mode) *m*/*z* 139 [M + H]^+^.*2-O-Caffeoylglycerol* (**4**): white amorphous powder; UV (MeOH) λ_max_ (log ε) 217 (3.65), 240 (3.25), 295 (sh), 325 (3.95) nm; ^1^H-NMR (700 MHz, CD_3_OD) δ 7.60 (1H, d, *J* = 16.1 Hz, H-7), 7.05 (1H, d, *J* = 2.1 Hz, H-2), 6.95 (1H, dd, *J* = 8.4, 2.1 Hz, H-6), 6.78 (1H, d, *J* = 8.4 Hz, H-5), 6.31 (1H, d, *J* = 16.1 Hz, H-8), 4.98 (1H, p, *J* = 5.6 Hz, H-2′), 3.76 (2H, dd, *J* = 11.9, 4.9 Hz, H_a_-1′, 3′), 3.73 (2H, dd, *J* = 11.9, 5.6 Hz, H_b_-1′, 3′); ^13^C-NMR (175 MHz, CD_3_OD) δ 169.1 (C-9), 149.7 (C-4), 147.2 (C-7), 147.0 (C-3), 127.9 (C-1), 123.1 (C-6), 116.6 (C-5), 115.5 (C-8), 115.3 (C-2), 76.7 (C-2′), 61.9 (C-1′, 3′); ESI-MS (positive ion mode) *m*/*z* 255 [M + H]^+^.*1-O-Caffeoylglycerol* (**5**): white amorphous powder; UV (MeOH) λ_max_ (log ε) 217 (3.68), 241 (3.29), 295 (sh), 325 (3.97) nm; ^1^H-NMR (700 MHz, CD_3_OD) δ 7.59 (1H, d, *J* = 15.4 Hz, H-7), 7.05 (1H, d, *J* = 2.1 Hz, H-2), 6.95 (1H, dd, *J* = 8.4, 2.1 Hz, H-6), 6.78 (1H, d, *J* = 2.1 Hz, H-5), 6.29 (1H, d, *J* = 15.4 Hz, H-8), 4.25 (1H, dd, *J* = 11.9, 4.2 Hz, H_a_-1′), 4.16 (1H, dd, *J* = 11.9, 4.2 Hz, H_b_-1′), 3.89 (1H, p, *J* = 6.3, 4.2 Hz, H-2′), 3.61 (1H, dd, *J* = 11.2, 4.9 Hz, H_a_-3′), 3.59 (1H, dd, *J* = 11.2, 5.6 Hz, H_b_-3′); ^13^C-NMR (175 MHz, CD_3_OD) δ 169.4 (C-9), 149.7 (C-4), 147.3 (C-7), 146.9 (C-3), 127.9 (C-1), 123.1 (C-6), 116.6 (C-5), 115.3 (C-2), 115.0 (C-8), 71.4 (C-2′), 66.7 (C-1′), 64.2 (C-3′); ESI-MS (positive ion mode) *m*/*z* 255 [M + H]^+^.*Juncusyl ester B* (**6**): white amorphous powder; UV (MeOH) λ_max_ (log ε) 210 (3.45), 226 (3.55), 297 (sh), 309 (4.05) nm; ^1^H-NMR (700 MHz, CD_3_OD) δ 7.67 (1H, d, *J* = 16.1 Hz, H-7), 7.46 (2H, d, *J* = 8.4 Hz, H-2, 6), 6.81 (2H, d, *J* = 8.4 Hz, H-3, 5), 6.37 (1H, d, *J* = 16.1 Hz, H-8), 4.99 (1H, p, *J* = 4.2 Hz, H-2′), 3.76 (1H, dd, *J* = 11.9, 4.9 Hz, H_a_-1′, 3′), 3.73 (1H, dd, *J* = 11.9, 5.6 Hz, H_b_-1′, 3′); ^13^C-NMR (175 MHz, CD_3_OD) δ 169.1 (C-9), 161.4 (C-4), 146.8 (C-7), 131.3 (C-2, 6), 127.4 (C-1), 117.0 (C-3, 5), 115.5 (C-8), 76.8 (C-2′), 61.9 (C-1′, 3′); ESI-MS (positive ion mode) *m*/*z* 239 [M + H]^+^.*Acertannin* (**7**): white amorphous powder; UV (MeOH) λ_max_ (log ε) 217 (4.15), 275 (3.53) nm; ^1^H-NMR (700 MHz, CD_3_OD) δ 7.09 (2H, s, H-2″, 6″), 7.08 (2H, s, H-2′, 6′), 4.91 (1H, ddd, *J* = 10.5, 9.1, 5.6 Hz, H-2), 4.55 (1H, dd, *J* = 12.6, 2.1 Hz, H_a_-6), 4.38 (1H, dd, *J* = 11.2, 4.9 Hz, H_b_-6), 4.10 (1H, dd, *J* = 11.2, 5.6 Hz, H_a_-1), 3.71 (1H, t, *J* = 9.1 Hz, H-3), 3.53 (1H, m, H-5), 3.52 (1H, m, H-4), 3.35 (1H, t-like, *J* = 10.5 Hz, H_b_-1); ^13^C-NMR (175 MHz, CD_3_OD) δ 168.5 (C-7″), 167.9 (C-7′), 146.7 (C-3″, 5″), 146.6 (C-3′, 5′), 140.1 (C-4′), 140.0 (C-4″), 121.5 (C-1″), 121.3 (C-1′), 110.4 (C-2′, 6′), 110.3 (C-2″, 6″), 80.3 (C-5), 77.1 (C-3), 73.3 (C-2), 72.1 (C-4), 68.1 (C-1), 65.0 (C-6); ESI-MS (positive ion mode) *m*/*z* 469 [M + H]^+^.*2-O-(E)-Feruloyl glyceride* (**8**): white amorphous powder; UV (MeOH) λ_max_ (log ε) 217 (3.60), 237 (2.50), 296 (sh), 325 (3.95) nm; ^1^H-NMR (700 MHz, CD_3_OD) δ 7.66 (1H, d, *J* = 16.1 Hz, H-7), 7.19 (1H, d, *J* = 2.1 Hz, H-2), 7.08 (1H, dd, *J* = 8.4, 2.1 Hz, H-6), 6.81 (1H, d, *J* = 8.4 Hz, H-5), 6.40 (1H, d, *J* = 16.1 Hz, H-8), 4.99 (1H, p, *J* = 4.9 Hz, H-2′), 3.89 (3H, s, 3-OCH_3_), 3.76 (1H, dd, *J* = 11.9, 4.2 Hz, H_a_-1′, 3′), 3.73 (1H, dd, *J* = 11.9, 5.6 Hz, H_b_-1′, 3′); ^13^C-NMR (175 MHz, CD_3_OD) δ 169.1 (C-9), 150.7 (C-4), 149.5 (C-3), 147.1 (C-7), 127.9 (C-1), 124.2 (C-6), 116.6 (C-5), 115.5 (C-8), 111.9 (C-2), 76.8 (C-2′), 61.9 (C-1′, 3′), 56.6 (3-OCH_3_); ESI-MS (positive ion mode) *m*/*z* 269 [M + H]^+^.*Maplexin D* (**9**): white amorphous powder; UV (MeOH) λ_max_ (log ε) 216 (4.10), 276 (3.47) nm; ^1^H-NMR (700 MHz, CD_3_OD) δ 7.10 (2H, s, H-2″, 6″), 7.08 (2H, s, H-2′, 6′), 4.99 (1H, ddd, *J* = 10.5, 9.1, 5.6 Hz, H-2), 5.01 (1H, t, *J* = 9.1 Hz, H-4), 4.18 (1H, dd, *J* = 11.2, 5.6 Hz, H_a_-1), 3.98 (1H, t, *J* = 9.1 Hz, H-3), 3.62 (1H, brd, *J* = 9.8 Hz, H_a_-6), 3.52 (1H, m, H_b_-6), 3.54 (1H, m, H-5), 3.40 (1H, t, *J* = 11.2 Hz, H_b_-1); ^13^C-NMR (175 MHz, CD_3_OD) δ 167.78 (C-7″), 167.82 (C-7′), 146.69 (C-3′, 5′), 146.65 (C-3″, 5″), 140.22 (C-4″), 140.18 (C-4′), 121.23 (C-1″), 121.2 (C-1′), 110.5 (C-2″, 6″), 110.4 (C-2′, 6′), 81.2 (C-5), 75.1 (C-3), 73.5 (C-2), 73.0 (C-4), 68.1 (C-1), 62.8 (C-6); ESI-MS (positive ion mode) *m*/*z* 469 [M + H]^+^.*trans-Caffeic acid* (**10**): white amorphous powder; UV (MeOH) λ_max_ (log ε) 217 (3.62), 239 (3.20), 297 (sh), 324 (3.92) nm; ^1^H-NMR (700 MHz, CD_3_OD) δ 7.53 (1H, d, *J* = 16.1 Hz, H-7), 7.03 (1H, d, *J* = 2.1 Hz, H-2), 6.93 (1H, dd, *J* = 8.4, 2.1 Hz, H-6), 6.77 (1H, d, *J* = 8.4 Hz, H-5), 6.21 (1H, d, *J* = 16.1 Hz, H-8); ^13^C-NMR (175 MHz, CD_3_OD) δ 171.2 (C-9), 149.6 (C-4), 147.2 (C-7), 147.0 (C-3), 127.9 (C-1), 123.0 (C-6), 116.6 (C-5), 115.2 (C-2), 115.7 (C-8); ESI-MS (positive ion mode) *m*/*z* 181 [M + H]^+^.*Syringic acid* (**11**): white amorphous powder; UV (MeOH) λ_max_ (log ε) 217 (4.15), 275 (3.42) nm; ^1^H-NMR (700 MHz, CD_3_OD) δ 7.33 (2H, s, H-2, 6), 3.88 (6H, s, 3, 5-OCH_3_); ^13^C-NMR (175 MHz, CD_3_OD) δ 170.1 (C-7), 149.0 (C-3, 5), 141.9 (C-4), 122.0 (C-1), 108.4 (C-2, 6), 56.9 (3, 5-OCH_3_); ESI-MS (positive ion mode) *m*/*z* 199 [M + H]^+^.*1-Syringoyl-1,2-dihydroxyethane* (**12**): white amorphous powder; UV (MeOH) λ_max_ (log ε) 205 (3.87), 230 (sh), 304 (3.35) nm; ^1^H-NMR (700 MHz, CD_3_OD) δ 7.34 (2H, s, H-2, 6), 5.14 (1H, dd, *J* = 5.6, 4.2 Hz, H-8), 3.91 (6H, s, 3, 5-OCH_3_), 3.89 (1H, dd, *J* = 11.2, 4.2 Hz, H_a_-9), 3.75 (1H, dd, *J* = 11.2, 4.2 Hz, H_b_-9); ^13^C-NMR (175 MHz, CD_3_OD) δ 199.9 (C-7), 149.3 (C-3, 5), 143.2 (C-4), 126.9 (C-1), 107.9 (C-2, 6), 75.7 (C-8), 66.4 (C-9), 57.1 (3, 5-OCH_3_); ESI-MS (positive ion mode) *m*/*z* 243 [M + H]^+^.*3,5-Dihydroxy-4-methoxybenzoic acid* (**13**): white amorphous powder; UV (MeOH) λ_max_ (log ε) 211 (3.97), 258 (3.39), 294 (3.15) nm; ^1^H-NMR (700 MHz, CD_3_OD) δ 7.03 (2H, s, H-2, 6), 3.89 (3H, s, 4-OCH_3_); ^13^C-NMR (175 MHz, CD_3_OD) δ 170.0 (C-7), 151.8 (C-3, 5), 141.2 (C-4), 127.3 (C-1), 110.5 (C-2, 6), 60.9 (4-OCH_3_); ESI-MS (positive ion mode) *m*/*z* 185 [M + H]^+^.*1-(3,4,5-Trimethoxyphenyl)-1,2,3-propanetriol* (**14**): white amorphous powder; UV (MeOH) λ_max_ (log ε) 204 (3.85), 227 (sh), 269 (2.55) nm; ^1^H-NMR (700 MHz, CD_3_OD) δ 6.72 (2H, s, H-2, 6), 4.59 (1H, d, *J* = 5.6 Hz, H-7), 3.84 (6H, s, 3, 5-OCH_3_), 3.74 (3H, s, 4-OCH_3_), 3.67 (1H, m, H-8), 3.54 (1H, dd, *J* = 11.2, 4.2 Hz, H_a_-9), 3.40 (1H, dd, *J* = 11.2, 5.6 Hz, H_b_-9); ^13^C-NMR (175 MHz, CD_3_OD) δ 154.5 (C-3, 5), 139.9 (C-1), 138.4 (C-4), 105.2 (C-2, 6), 77.5 (C-8), 75.4 (C-7), 64.4 (C-9), 61.2 (4-OCH3), 56.7 (3, 5-OCH_3_); ESI-MS (positive ion mode) *m*/*z* 259 [M + H]^+^.*1-O-trans-ρ-Coumaroylglycerol* (**15**): white amorphous powder; UV (MeOH) λ_max_ (log ε) 210 (3.35), 226 (3.45), 296 (sh), 310 (3.90) nm; ^1^H-NMR (700 MHz, CD_3_OD) δ 7.65 (1H, d, *J* = 16.1 Hz, H-7), 7.46 (1H, d, *J* = 9.1 Hz, H-2, 6), 6.80 (1H, d, *J* = 9.1 Hz, H-3, 5), 6.36 (1H, d, *J* = 16.1 Hz, H-8), 4.26 (1H, dd, *J* = 11.2, 4.2 Hz, H_a_-1′), 4.16 (1H, dd, *J* = 11.2, 6.3 Hz, H_b_-1′), 3.89 (1H, m, H-2′), 3.61 (1H, dd, *J* = 11.2, 5.6 Hz, H_a_-3′), 3.59 (1H, dd, *J* = 11.2, 5.6 Hz, H_b_-3′); ^13^C-NMR (175 MHz, CD_3_OD) δ 169.3 (C-9), 161.5 (C-4), 146.9 (C-7), 131.3 (C-2, 6), 127.3 (C-1), 117.0 (C-3, 5), 115.1 (C-8), 71.4 (C-2′), 66.7 (C-1′), 64.2 (C-3′); ESI-MS (positive ion mode) *m*/*z* 239 [M + H]^+^.*Sieboldic acid* (**16**): white amorphous powder; ${\left[\alpha \right]}_{D}^{21}$ −4.8° (*c* = 0.14, MeOH); ^1^H-NMR (700 MHz, CD_3_OD) and ^13^C-NMR (175 MHz, CD_3_OD) (Table 1); HRESI-MS (positive ion mode) *m*/*z* 199.0575 [M + Na]^+^ (calcd. for C_7_H_12_O_5_Na, 199.0577).*Smilaxsiebolane A* (**17**): white amorphous powder; ${\left[\alpha \right]}_{D}^{25}$ +46.4° (*c* = 0.11, CH_2_Cl_2_); IR (KBr) ν_max_ 3382, 3043, 2920, 2856, 1706, 1634, 1462, 1371, 1044, 979, 890 cm^−1^; ^1^H-NMR (700 MHz, CDC1_3_) and ^13^C-NMR (175 MHz, CDC1_3_) (Table 2); HRESI-MS (positive ion mode) *m*/*z* 455.4275 [M + H]^+^ (calcd. for C_32_H_55_O, 455.4247).*Smilaxsiebolane B* (**18**): white amorphous powder; ${\left[\alpha \right]}_{D}^{25}$ +26.4° (*c* = 0.36, CH_2_Cl_2_); IR (KBr) ν_max_ 2959, 2870, 1706, 1635, 1461, 1368, 1114, 986, 890 cm^−1^; ^1^H-NMR (700 MHz, CDC1_3_) and ^13^C-NMR (175 MHz, CDC1_3_) (Table 2); HRESI-MS (positive ion mode) *m*/*z* 453.4071 [M + H]^+^ (calcd. for C_32_H_53_O, 453.4091).*trans-ρ-Ethyl coumarate* (**19**): white amorphous powder; UV (MeOH) λ_max_ (log ε) 217 (3.44), 236 (3.53), 294 (sh), 325 (4.13) nm; ^1^H-NMR (700 MHz, CD_3_OD) δ 7.60 (1H, d, *J* = 16.1 Hz, H-7), 7.45 (1H, d, *J* = 8.4 Hz, H-2, 6), 6.80 (1H, d, *J* = 8.4 Hz, H-3, 5), 6.31 (1H, d, *J* = 16.1 Hz, H-8), 4.21 (2H, q, *J* = 7.0 Hz, H-1′), 1.31 (1H, t, *J* = 7.0 Hz, H-2′); ^13^C-NMR (175 MHz, CD_3_OD) δ 169.5 (C-9), 161.4 (C-4), 146.5 (C-7), 131.3 (C-2, 6), 127.3 (C-1), 117.0 (C-3, 5), 115.5 (C-8), 61.6 (C-1′), 14.8 (C-2′); ESI-MS (positive ion mode) *m*/*z* 193 [M + H]^+^.

### 3.5. Cell Culture and Adipocyte Differentiation

The mouse 3T3-L1 preadipocytes were procured from Zenbio (SP-L1-F, Zenbio, Durham, NC, USA), and cells were cultured in 3T3-L1 Preadipocyte Medium (PM-1-L1, Zenbio) in a humidified incubator with 95% air and 5% CO_2_ at 37 °C. Two days after cells reached confluence, induction with initiation media [3T3-L1 Differentiation Medium (3-isobutyl-1-methylxanthine, dexamethasone, and insulin (MDI)); DM-2-L1, Zenbio] was performed. The initiation medium was changed on day two with the 3T3-L1 Adipocyte Medium (AM-1-L1, Zenbio). On days four and six, the progression medium was replaced with the 3T3-L1 Adipocyte Medium (AM-1-L1, Zenbio). From days zero to eight, the cells were treated with nontoxic concentrations of the extracts and isolated constituents, which were identified via a cell cytotoxicity assay.

### 3.6. Oil Red O Staining of Adipocytes Lipid Droplets

On day eight, following differentiation induction, differentiated 3T3-L1 cells were stained with a lipid (Oil Red O) staining kit (ST-R100, Zenbio). Briefly, the cells were washed with PBS, fixed with a fixation solution (Zenbio) for 30 min in the dark, and washed two times with PBS and 70% EtOH. The lipid droplets within the differentiated 3T3-L1 cells were then stained with an Oil Red O solution (Zenbio) for 30 min. The excess stain was removed by washing with 70% EtOH and PBS. The stained lipid droplets were dissolved in isopropanol containing 4% nonidet P-40 (Sigma-Aldrich, St. Louis, MO, USA), and quantitative analysis using an enzyme-linked immunosorbent assay (ELISA) reader (SPECTRAmax 190PC, Molecular Devices, San Jose, CA, USA) at 510 nm was performed.

### 3.7. Cell Viability Assay

The cell viability was analyzed by MTT cytotoxicity assays [11]. The 3T3-L1 preadipocytes were seeded at a density of 5 × 10^3^ cells/well in 100 μL of culture medium. A time-zero control plate was prepared one day after plating. The isolated compounds (**1**–**19**) were applied directly, and the cells were incubated for 24 h in a humidified atmosphere with 5% CO_2_ at 37 °C. The proliferation of the cells was then determined. MTT [5 mg/mL in phosphate-buffered saline (PBS)] solution was added to the wells, followed by incubation for 3 h. The medium was removed from the wells by aspiration, and buffered dimethyl sulfoxide (DMSO, 0.1 mL) was added to each well, after which the plates were shaken. Subsequently, absorbance was measured using a SpectraMax 190PC microtiter plate reader at 540 nm.

### 3.8. Statistical Analysis

The presentation of all data is as means standard deviation (SD). All experiments were performed at least three times independently. Differences between the means of each experimental group were analyzed using a one-way analysis of variance (ANOVA) followed by Tukey’s post-hoc test (IBM SPSS statistics 29). Values were considered statistically significant differences, which means sharing the different superscript letters when *p* < 0.05. ANOVA and Student’s *t*-test were used to determine the significance of the results. Values of * *p* < 0.05 and ** *p* < 0.01 were considered statistically significant.

## 4. Conclusions

In this study, as part of an ongoing research project to discover bioactive natural products, we identified anti-adipogenic secondary metabolites from the whole plant of the ethanolic extracts of *S. sieboldii* that inhibit adipocyte differentiation in 3T3-L1 cells. Our biological data for the first time revealed that stilbenoids, phenylpropanoid glycerides, and gallotanins might be responsible for the reported inhibitory activities on adipocyte differentiation in 3T3-L1 cells of the *S. sieboldii* extracts. Therefore, we conclude that the anti-obesity effects of *S. sieboldii* extracts are mediated via anti-adipogenesis. The bioactive constituents with anti-adipogenic activity, 2-*O*-caffeoylglycerol (**4**), acertannin (**7**), maplexin D (**9**), and *trans*-ρ-ethyl coumarate (**19**) can be further chemically optimized to obtain the additional active compounds for the inhibition of adipocyte differentiation. In addition, the active compounds can be further studied for their exact underlying mechanisms, and these can also be investigated in obesity models in vivo for establishing therapeutic potential. Taken together, our findings suggest that the regulation of adipocyte differentiation is possible, thereby making *S. sieboldii* a promising source for the development of therapeutic agents and health-promoting components to treat diseases associated with obesity.

## Figures and Tables

**Figure 1 ijms-24-08866-f001:**
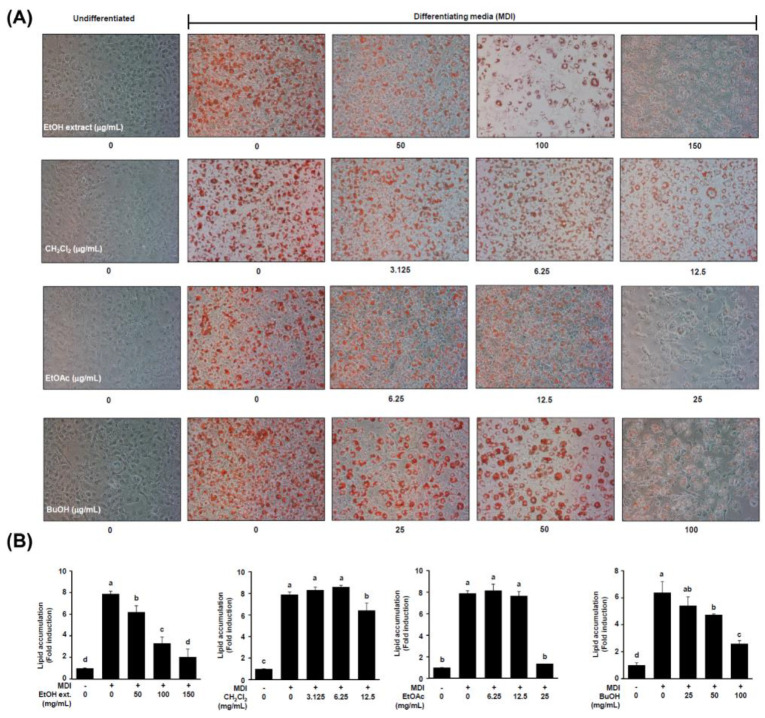
Anti-adipogenic effects of *S. sieboldii* ethanol (EtOH) extract and three solvent fractions. 3T3-L1 cells were treated with varied concentrations of crude EtOH extract and solvent fraction from start of adipocyte differentiation (designated day 0) for eight days. (**A**) The representative Oil Red O staining pictures of differentiated adipocytes after eight days. (**B**) Quantitative analysis of Oil Red O staining levels. Values are represented as means ± SD of 3 experiments. Differences among groups were determined by one-way ANOVA followed by Tukey’s post-hoc test; the different superscript letters (a–d) are significantly different from each group (*p* < 0.05). MDI: 3-isobutyl-1-methylxanthine (IBMX), dexamethasone (DEX), and insulin.

**Figure 2 ijms-24-08866-f002:**
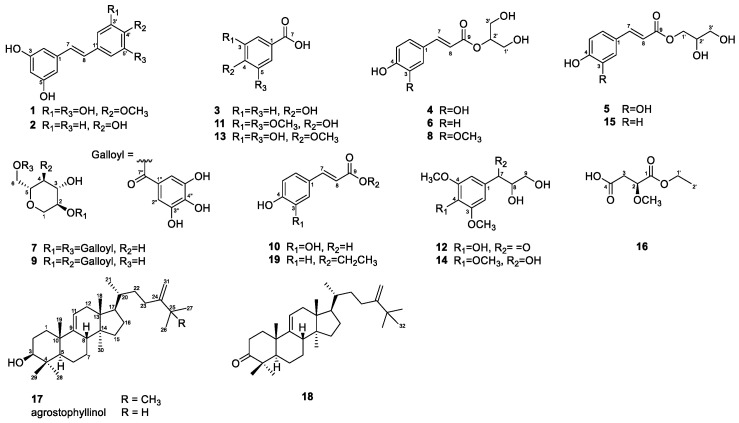
Chemical structures of compounds **1**–**19**.

**Figure 3 ijms-24-08866-f003:**
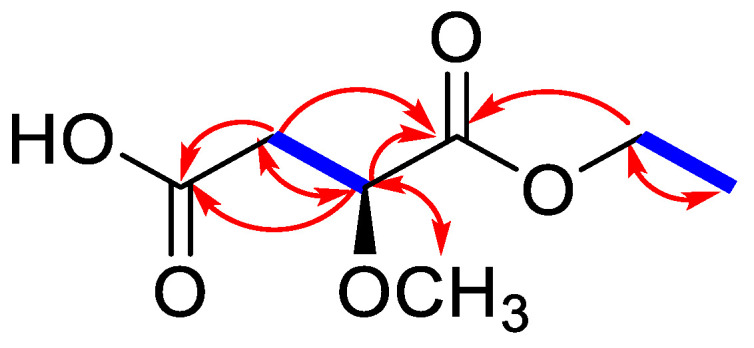
Key ^1^H-^1^H COSY (bold blue lines) and HMBC (red arrows) correlations of **16**.

**Figure 4 ijms-24-08866-f004:**
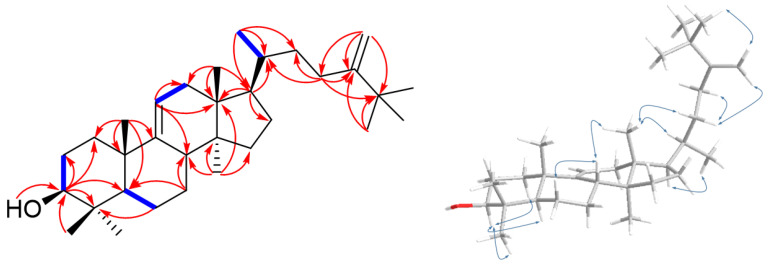
Key ^1^H-^1^H COSY (blue bold lines), HMBC (red arrows), and ROESY (blue arrow) correlations of **17**.

**Figure 5 ijms-24-08866-f005:**
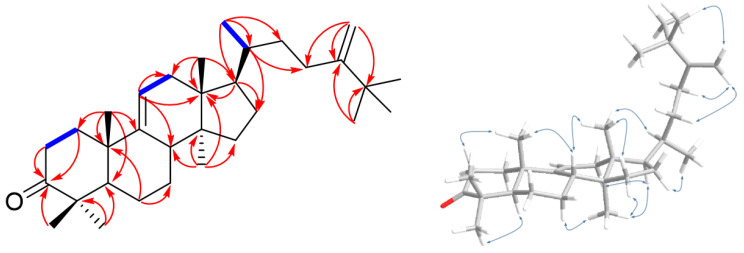
Key ^1^H-^1^H COSY (blue bold lines), HMBC (red arrows), and ROESY (blue arrow) correlations of **18**.

**Figure 6 ijms-24-08866-f006:**
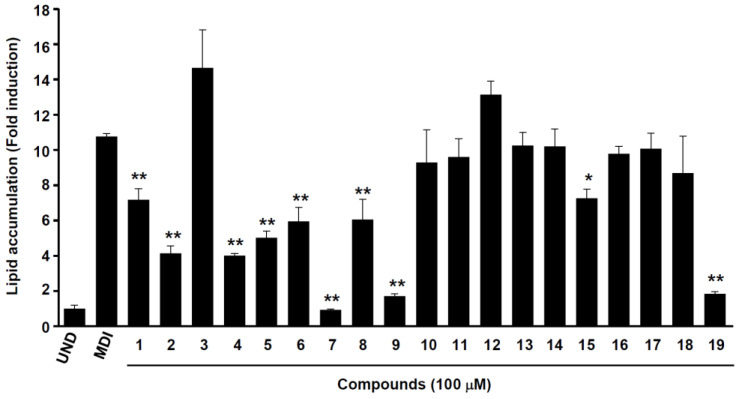
Anti-adipogenic activity of isolated constituents from whole plant of *S. sieboldii* CH_2_Cl_2_- and EtOAc-soluble extracts. UND: undifferentiated, one-way analysis of variance (ANOVA) and Student’s *t*-test were used to determine the significance of the results. Values of * *p* < 0.05 and ** *p* < 0.01 were considered statistically significant (compared with MDI-treated cells). MDI: 3-isobutyl-1-methylxanthine (IBMX), dexamethasone (DEX), and insulin.

**Figure 7 ijms-24-08866-f007:**
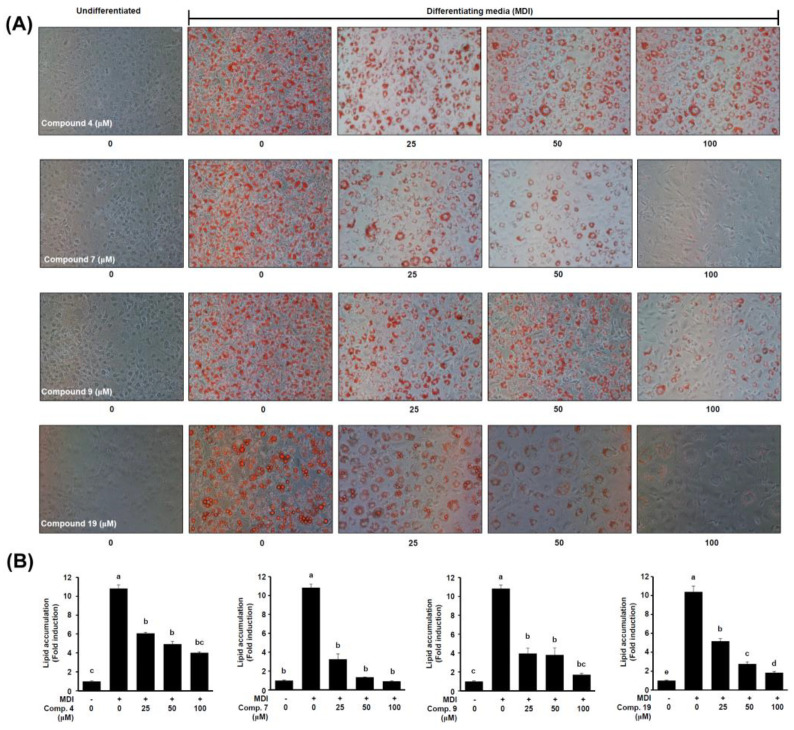
Dosage-dependent research of active compounds, 2-*O*-caffeoylglycerol (**4**), acertannin (**7**), maplexin D (**9**), and *trans*-ρ-ethyl coumarate (**19**) on lipid accumulation in adipocytes. (**A**) The representative Oil Red O staining pictures of differentiated adipocytes on day 8. (**B**) Quantitative analysis of Oil Red O staining contents. Differences among groups were determined by one-way ANOVA followed by Tukey’s post-hoc test; the different superscript letters (a–e) are significantly different from each group (*p* < 0.05). MDI: 3-isobutyl-1-methylxanthine (IBMX), dexamethasone (DEX), and insulin.

**Figure 8 ijms-24-08866-f008:**
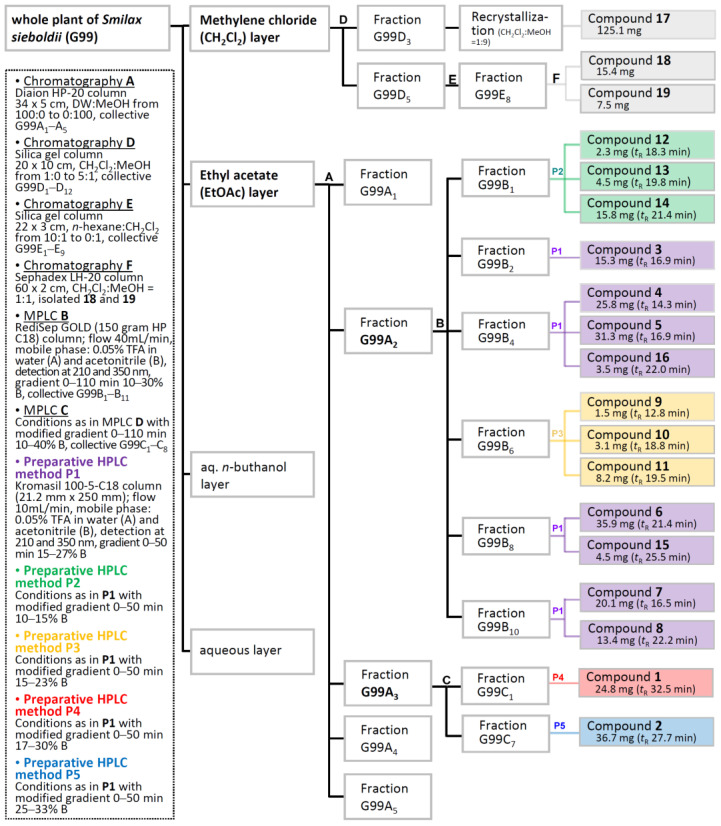
The isolation scheme of compounds **1**–**19**.

**Table 1 ijms-24-08866-t001:** ^1^H and ^13^C NMR spectroscopic data for compound **16** in CD_3_OD (δ in ppm) *^a^*.

Position	16	
	δ_H_ (*J* in Hz)	δ_C_ (mult.)
1		173.5 s
2	4.16 dd (8.4, 4.2)	78.6 d
3	2.71 dd (16.1, 4.2); 2.60 dd (16.1, 8.4)	39.4 t
4		174.7 s
1′	4.22 dd (7.0, 2.8); 4.20 dd (7.0, 3.5)	62.4 t
2′	1.28 t (7.0)	14.6 q
OCH_3_	3.41 s	59.0 q

*^a^* Assignments confirmed by ^1^H-^1^H COSY, HSQC, and HMBC experiments.

**Table 2 ijms-24-08866-t002:** ^1^H and ^13^C NMR spectroscopic data for compounds **17** and **18** in CDCl_3_ (δ in ppm) *^a^*.

Position	17		18	
	δ_H_ (*J* in Hz)	δ_C_ (mult.)	δ_H_ (*J* in Hz)	δ_C_ (mult.)
1α	1.42 m	36.3 t	2.10 m	36.7 t
1β	1.77 td (13.3, 3.5)		1.81 dt (13.3, 4.9)	
2α	1.74 m	28.0 t	2.40 m	34.9 t
2β	1.60 m		2.72 m	
3α	3.20 dd (11.2, 4.2)	79.1 d		217.2 s
4		39.3 s		47.7 s
5α	0.86 m	52.7 d	1.37 m	53.4 d
6α	1.68 m	21.6 t	1.64 m	22.6 t
6β	1.45 dd (13.3, 3.5)			
7α	1.28 m	28.3 t	1.36 m	27.7 t
7β	1.65 m		1.70 m	
8β	2.16 m	42.0 d	2.22 m	41.9 d
9		148.7 s		147.1 s
10		39.6 s		39.1 s
11	5.21 d (5.6)	115.2 d	5.29 d (6.3)	116.3 d
12α	2.06 m	37.4 t	2.09 m	37.2 t
12β	1.88 m		1.93 m	
13		44.5 s		44.3 s
14		47.2 s		47.0 s
15α	1.30 m	34.1 t	1.33 m	33.9 t
15β	1.37 m		1.40 m	
16α	1.90 m	28.2 t	1.95 m	28.0 t
16β	1.27 m		1.31 m	
17α	1.61 m	51.2 d	1.65 m	50.9 d
18	0.64 s	14.6 q	0.68 s	14.4 q
19	1.03 s	22.5 q	1.23 s	21.8 q
20β	1.39 m	36.71 d	1.43 m	36.5 d
21	0.90 d (6.3)	18.73 q	0.93 d (6.3)	18.5 q
22α	1.14 m	36.69 t	1.17 m	36.4 t
22β	1.56 m		1.59 m	
23α	2.13 m	28.4 t	2.15 m	28.2 t
23β	1.85 m		1.89 m	
24		159.2 s		159.0 s
25		36.5 s		36.3 s
26	1.04 s	29.6 q	1.06 s	29.3 q
27	1.04 s	29.6 q	1.06 s	29.3 q
28	0.97 s	28.5 q	1.08 s	25.6 q
29	0.80 s	15.9 q	1.07 s	22.0 q
30	0.73 s	18.71 q	0.75 s	18.4 q
31	4.82 s, 4.65 s	105.9 t	4.84 s, 4.67 s	105.8 t
32	1.04 s	29.6 q	1.06 s	29.3 q

*^a^* Measured at 700 and 175 MH. The assignments were based on ^1^H-^1^H COSY, HSQC, HMBC, and ROESY experiments.

## Data Availability

Data is contained within the article and Appendix A.

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
