# Peer review of "Anti-Adipogenic Activity of Secondary Metabolites Isolated from *Smilax sieboldii* Miq. on 3T3-L1 Adipocytes"

_ijms, 2023, doi:10.3390/ijms24108866_

Round 1
Reviewer 1 Report
The current manuscript is an interesting and robust study regarding the anti-adipogenic activity of Smilax sieboldii compounds. Nevertheless, some alterations should be made before acceptance for publication:
- Article type should be “Article” and not “Communication” (since it is an experimental study);
- The abstract should contain at least a small sentence on what is Smilax sieboldii, as an introduction;
- The introduction section should include current therapies and/or methods (Nutritional? Exercise?) that are usually applied to fight obesity;
- An image should be included regarding obesity pathophysiology, causes and consequences, and current therapies and respective limitations;
- Figures and Tables should be placed immediately after they are first mentioned in the text;
- Figure 1 quality (resolution) should be improved, suggest making images bigger;
- Abbreviations present in Figures and Tables should be defined in the respective caption, independently of having already been defined in the text;
- Figures 6 and 7 should be included in the Results section, not the Methods section;
- “Scheme 1” is a Figure, hence it should be called “Figure”, with the appropriate numbering;
- The software used for statistical analysis should be mentioned;
- The “Conclusions” section should be named “Final remarks” or “Discussion”, and a more summarized “Conclusion” section should be added afterwards;
- Therapeutic potential of the isolated compounds should be discussed more thoroughly, namely in what concerns possible administration route(s), and formulation(s) (solid dosage forms, liquid, semisolid? Conventional or nanosystems?).
Author Response
Responses to Referee
Referee 1
Comments and Suggestions for Authors
The current manuscript is an interesting and robust study regarding the anti-adipogenic activity of Smilax sieboldii compounds. Nevertheless, some alterations should be made before acceptance for publication:
- Article type should be “Article” and not “Communication” (since it is an experimental study);
▷ It was modified. (page 1, line 1)
- The abstract should contain at least a small sentence on what is Smilax sieboldii, as an introduction;
▷ It was modified. (page 1, line 13 ~ 14)
- The introduction section should include current therapies and/or methods (Nutritional? Exercise?) that are usually applied to fight obesity;
▷ It was modified. (page 1, line 40 ~ 42)
- An image should be included regarding obesity pathophysiology, causes and consequences, and current therapies and respective limitations;
▷ It was modified. (page 1, line 43 page 2, line 49)
- Figures and Tables should be placed immediately after they are first mentioned in the text;
▷ Relocate Figures and Tables.
- Figure 1 quality (resolution) should be improved, suggest making images bigger;
▷ Replace image (Figures 1 and 7).
- Abbreviations present in Figures and Tables should be defined in the respective caption, independently of having already been defined in the text;
▷ It was modified. (page 8, line 243 ~ 244; page 9, line 278 ~ 279)
- Figures 6 and 7 should be included in the Results section, not the Methods section;
▷ Relocate Figures 6 and 7.
- “Scheme 1” is a Figure, hence it should be called “Figure”, with the appropriate numbering;
▷ It was modified. (page 11, line 354; Figure 8)
- The software used for statistical analysis should be mentioned;
▷ It was modified. (page 14, line 490)
- The “Conclusions” section should be named “Final remarks” or “Discussion”, and a more summarized “Conclusion” section should be added afterwards;
▷ It was modified. (page 14, line 495~510)
- Therapeutic potential of the isolated compounds should be discussed more thoroughly, namely in what concerns possible administration route(s), and formulation(s) (solid dosage forms, liquid, semisolid? Conventional or nano systems?).
▷ It was modified. (page 14, line 505~507)

Reviewer 2 Report
The manuscript entitled "Anti-Adipogenic Activity of Secondary Metabolites Isolated from Smilax sieboldii Miq. on 3T3-L1 Adipocytes” evaluates the biological anti-adipogenic activity of methylene chloride, ethyl acetate, aqueous-saturated n-butanol, and ethanol extracts of Smilax sieboldii on the 3T3-L1 pre-adipocytes. The extraction methodology and chemical identification are well-characterized. The preliminary biological studies included the differentiation of the preadipocytes into the mature form using the Oil 100 Red O staining. The conclusions are appropriate and are based on study results.
The manuscript contains a set of useful data, both from a chemical and biological point of view, however, some improvements are necessary for the manuscript.
Comments:
1) Why did not the authors attempt to address preliminary mechanisms related to adipogenesis such as the expression of adipogenic and lipogenic proteins, ROS production, and MAPK signaling? Moreover, no mechanisms related to lipogenesis such as glucose uptake into adipocytes.
2) In section 2.3. (Anti-Adipogenic Effects of Isolated Compounds on 3T3-L1 Cells). Why was incubation time (24 h) used in the manuscript? Treatment for only 24 h seems not able to give enough picture of the effect of isolated compounds on cell viability. In general, a time course covering up to 24 or 72 h should be expected.
3) Similar to the isolated compounds, the authors should assess the effect of the whole extract on the viability of 3T3-L1 Cells.
4) To clarify it for readers, in Figure 1, how did the authors decide on using 50-150 µg/mL for the tested extracts? The authors are advised to address this point and add the answers/proper citations to the relevant results section.
5) To clarify it for readers, in Figure 6, how did the authors decide on using 100 µM concentration for the tested extracts? The authors are advised to address this point and add the answers/proper citations to the relevant results section.
6) The authors are advised to describe the rationale for using resveratrol as the positive control in the antiadipogenic experiment. Please, elaborate on this point and describe the implicated mechanisms for this anti-adipogenic effect.
7) In the statistical analysis (section 3.8.), did the authors check data normality and homogeneity of data before proceeding to one-way ANOVA or t-test which are valid only for the analysis of normally distributed data? Authors are advised to address this point and add the answers to the comment in the material and methods section.
8) Since the study involves several experimental treatments, statistical analysis is typically analyzed by ANOVA followed by a post-hoc test e.g., Tukey-Kramer. The combination of t-test and one-way ANOVA might not be appropriate. The authors are advised to redo the statistical analysis as described above.
9) Some typo/syntax errors are present in the manuscript which need to be addressed, for example:
- In line 436, the authors state “3.5. Cell Culture and Adipocyte Ddifferentiation”. Please, correct “Differentiation” to correct “Ddifferentiation.
10) The authors are advised to carefully revise the reference section. The authors are advised to unify the way they write the journal name. Sometimes it is written as the full name (such as reference 21) while in other references it was written as an abbreviation. Please, follow the journal instructions in this regard.
Minor editing of English language required.
Author Response
Responses to Referee
Referee 2
The manuscript entitled "Anti-Adipogenic Activity of Secondary Metabolites Isolated from Smilax sieboldii Miq. on 3T3-L1 Adipocytes” evaluates the biological anti-adipogenic activity of methylene chloride, ethyl acetate, aqueous-saturated n-butanol, and ethanol extracts of Smilax sieboldii on the 3T3-L1 pre-adipocytes. The extraction methodology and chemical identification are well-characterized. The preliminary biological studies included the differentiation of the preadipocytes into the mature form using the Oil 100 Red O staining. The conclusions are appropriate and are based on study results.
The manuscript contains a set of useful data, both from a chemical and biological point of view, however, some improvements are necessary for the manuscript.
Comments:
1) Why did not the authors attempt to address preliminary mechanisms related to adipogenesis such as the expression of adipogenic and lipogenic proteins, ROS production, and MAPK signaling? Moreover, no mechanisms related to lipogenesis such as glucose uptake into adipocytes.
▷ You raised the important points, but the study on biological activities got to be conducted at the screening level because the study focused on the studies on the components by having views on the point that there are few studies on secondary metabolite of the research subject of the study in the process of finding functional natural materials. As mentioned in the conclusion, the study on mechanism will be conducted in the future.
2) In section 2.3. (Anti-Adipogenic Effects of Isolated Compounds on 3T3-L1 Cells). Why was incubation time (24 h) used in the manuscript? Treatment for only 24 h seems not able to give enough picture of the effect of isolated compounds on cell viability. In general, a time course covering up to 24 or 72 h should be expected.
▷ Thanks for your comment. As the reviewer's comment, cell viability is incubated for 24 hours generally. Therefore, 24 hours were applied to our experiment as well. However, for the next experiment, cell viability for 72 hours will also be experimented as pointed out by the reviewer (treatment for only 24 h seems not able to give enough picture of the effect of isolated compounds on cell viability).
3) Similar to the isolated compounds, the authors should assess the effect of the whole extract on the viability of 3T3-L1 Cells.
▷ According to reviewer's suggestion, the cell viability of the whole extract was confirmed in 3T3-L1 cells, and we also provide it as a supplementary data.
4) To clarify it for readers, in Figure 1, how did the authors decide on using 50-150 µg/mL for the tested extracts? The authors are advised to address this point and add the answers/proper citations to the relevant results section.
▷ Whole extract did not induce cytotoxicity below 150 µg/mL. Thus, the concentration ranges of whole extract selected treatment for adipocyte differentiation effect in 3T3-L1 cells were 50, 100, and 150 µg/mL. We provide references that can help explain our experimental results.
- Kang MJ, Kim KK, Son BY, Nam SW, Shin PG, Kim GD. The Anti-Adipogenic Activity of a New Cultivar, Pleurotus eryngii var. ferulae 'Beesan No. 2', through Down-Regulation of PPAR γ and C/EBP α in 3T3-L1 Cells. J Microbiol Biotechnol. 2016, 28;26(11):1836-1844.
- Hengpratom T, Ngernsoungnern A, Ngernsoungnern P, Lowe GM, Eumkeb G. Antiadipogenesis of Oroxylum indicum (L.) Kurz Extract via PPARγ2 in 3T3-L1 Adipocytes. Evid Based Complement Alternat Med. 2020, 5;2020:6720205.
5) To clarify it for readers, in Figure 6, how did the authors decide on using 100 µM concentration for the tested extracts? The authors are advised to address this point and add the answers/proper citations to the relevant results section.
▷ Generally, in the case of compounds isolated from the extract, experiments are performe by setting the maximum concentration to 100 or 200 uM. The compounds did not induce cytotoxicity below 100 μM. Therefore, we were decided the maximum concentration of compounds at 100 μM for treatment in 3T3-L1 cells. We provide references that can help explain our experimental results.
- Lee JA, Cho YR, Hong SS, Ahn EK. Anti-Obesity Activity of Saringosterol Isolated from Sargassum muticum (Yendo) Fensholt Extract in 3T3-L1 Cells. Phytother Res. 2017, 31(11):1694-1701.
- Aranaz P, Navarro-Herrera D, Zabala M, Miguéliz I, Romo-Hualde A, López-Yoldi M, Martínez JA, Vizmanos JL, Milagro FI, González-Navarro CJ. Phenolic Compounds Inhibit 3T3-L1 Adipogenesis Depending on the Stage of Differentiation and Their Binding Affinity to PPARγ. Molecules. 2019, 16;24(6):1045.
6) The authors are advised to describe the rationale for using resveratrol as the positive control in the antiadipogenic experiment. Please, elaborate on this point and describe the implicated mechanisms for this anti-adipogenic effect.
▷ It was modified. (page 8, line 260~264; add reference 54)
7) In the statistical analysis (section 3.8.), did the authors check data normality and homogeneity of data before proceeding to one-way ANOVA or t-test which are valid only for the analysis of normally distributed data? Authors are advised to address this point and add the answers to the comment in the material and methods section.
▷ According to reviewer's comment, we checked data normality and homogeneity and we added material and methods (3.8 statistical analysis) in the revised manuscript. Also we have modified figure legends.
à Figure legends (Fig. 1, 6, 7)
- page 3, line 104~106; Figure 1. Differences among groups were determined by one-way ANOVA followed by Tukey’s Post-hoc test; the different superscript letters (a-d) are significantly different from each group (P < 0.05).
- page 8, line 241~243; Figure 6. One-way analysis of variance (ANOVA) and Student's t-test and were used to determine the significance of the results. Values of *p < 0.05 and **p < 0.01 were considered statistically significant (compared with MDI-treated cells).
- page 9, line 275~277; Figure 7. Differences among groups were determined by one-way ANOVA followed by Tukey’s Post-hoc test; the different superscript letters (a-e) are significantly different from each group (P < 0.05).
- page 14, line 487~493; “3.8. Statistical Analysis”. The presentation of all data is as means standard deviation (SD). All experiments were performed at least three times independently. Differences between means of each experimental group were analyzed using one-way analysis of variance (ANOVA) fol-lowed by Tukey’s post hoc test (SPSS statistics 29). Values were considered statistically significant differences, which means sharing the different superscript letters when P < 0.05. ANOVA and Student's t-test and were used to determine the significance of the results. Values of *p < 0.05 and **p < 0.01 were considered statistically significant.
8) Since the study involves several experimental treatments, statistical analysis is typically analyzed by ANOVA followed by a post-hoc test e.g., Tukey-Kramer. The combination of t-test and one-way ANOVA might not be appropriate. The authors are advised to redo the statistical analysis as described above.
▷ According to reviewer's comment, we added material and methods (3.8 statistical analysis) in the revised manuscript and we have modified figure legends.
à Figure legends (Fig. 1, 6, 7)
- page 3, line 104~106; Figure 1. Differences among groups were determined by one-way ANOVA followed by Tukey’s Post-hoc test; the different superscript letters (a-d) are significantly different from each group (P < 0.05).
- page 8, line 241~243; Figure 6. One-way analysis of variance (ANOVA) and Student's t-test and were used to determine the significance of the results. Values of *p < 0.05 and **p < 0.01 were considered statistically significant (compared with MDI-treated cells).
- page 9, line 275~277; Figure 7. Differences among groups were determined by one-way ANOVA followed by Tukey’s Post-hoc test; the different superscript letters (a-e) are significantly different from each group (P < 0.05).
- page 14, line 487~493; “3.8. Statistical Analysis”. The presentation of all data is as means standard deviation (SD). All experiments were performed at least three times independently. Differences between means of each experimental group were analyzed using one-way analysis of variance (ANOVA) fol-lowed by Tukey’s post hoc test (SPSS statistics 29). Values were considered statistically significant differences, which means sharing the different superscript letters when P < 0.05. ANOVA and Student's t-test and were used to determine the significance of the results. Values of *p < 0.05 and **p < 0.01 were considered statistically significant.
9) Some typo/syntax errors are present in the manuscript which need to be addressed, for example:
- In line 436, the authors state “3.5. Cell Culture and Adipocyte Ddifferentiation”. Please, correct “Differentiation” to correct “Ddifferentiation.
▷ It was modified. (page 13, line 454)
10) The authors are advised to carefully revise the reference section. The authors are advised to unify the way they write the journal name. Sometimes it is written as the full name (such as reference 21) while in other references it was written as an abbreviation. Please, follow the journal instructions in this regard.
▷ It was modified. (page 13, line 454)
Editage (www.editage.co.kr, job code: RDCET_36_3) checked this manuscript for English.
